# The Geometry and Mechanics of Insect Wing Deformations in Flight: A Modelling Approach

**DOI:** 10.3390/insects11070446

**Published:** 2020-07-17

**Authors:** Robin Wootton

**Affiliations:** Department of Biosciences, University of Exeter, Address for correspondence 61 Thornton Hill, Exeter EX4 4NR, UK; r.j.wootton@exeter.ac.uk

**Keywords:** insects, wings, deformation, flight, bending, torsion, camber, control, physical models

## Abstract

The nature, occurrence, morphological basis and functions of insect wing deformation in flight are reviewed. The importance of relief in supporting the wing is stressed, and three types are recognized, namely corrugation, an M-shaped section and camber, all of which need to be overcome if wings are to bend usefully in the morphological upstroke. How this is achieved, and how bending, torsion and change in profile are mechanically interrelated, are explored by means of simple physical models which reflect situations that are visible in high speed photographs and films. The shapes of lines of transverse flexion are shown to reflect the timing and roles of bending, and their orientation is shown to determine the extent of the torsional component of the deformation process. Some configurations prove to allow two stable conditions, others to be monostable. The possibility of active remote control of wing rigidity by the thoracic musculature is considered, but the extent of this remains uncertain.

## 1. Introduction

This paper has a dual function: to review the occurrence of flight-related deformations in the morphological upstroke of insect wings and to investigate the geometric principles underlying the interaction of bending, torsion and camber change, by means of simple physical models.

Orthodox, flight-adapted insect wings are smart structures: they are flexible aerofoils whose three-dimensional shape from instant to instant in flight is largely determined by their elastic response to the aerodynamic and inertial forces they are receiving. While the profile of the wing base can normally be altered and controlled by the direct flight muscles of the thorax, the absence of muscles within the wing requires that three-dimensional shape control beyond the base is to a great extent automatic — encoded in the wing’s detailed structure. Four decades ago, I discussed the nature and function of the deformations they undergo, and identified a range of morphological adaptations to facilitate and to limit them [1]. The extensive research carried out since then has expanded and broadly confirmed these early conclusions and predictions [2,3,4,5,6,7,8,9,10,11,12,13,14,15,16,17,18,19,20,21,22,23,24,25,26] (in particular, see [19,21] for summaries of the extensive Russian literature), and major advances in insect aerodynamics have greatly helped to interpret their significance, e.g., [27,28,29,30,31,32,33,34,35].

Our knowledge of wing kinematics and deformations has come from high speed still and cine photography and video-recording. These sources show, unsurprisingly, that in the insects studied, the wings’ cyclic deformations are not rigidly determined: they vary in extent, even within a given flight sequence. To take just one example, high speed photographs of *Panorpa communis* (Mecoptera) in the upstroke published by Brackenbury [16] show virtually no bending in the wings, and Brodsky and Ivanov [4], filming tethered individuals, found little wing flexion, but a short high speed movie sequence of *Panorpa germanica* shortly after take-off shows extensive upstroke bending of the forewings and particularly the hindwings increasing from stroke to stroke [14] (Figure 1). These are different species, but their wings are structurally identical, and one would expect similar behaviour in both. These variations between strokes may be passive: wing shape must certainly be influenced by variations in angular velocity in the translation part of the stroke and in angular acceleration around stroke reversal. However, there is a possibility that, in some insects at least, a degree of control of bending, passive torsion and section may be exerted remotely by muscles at the wing base, and it is interesting to explore how such control might be achieved. Furthermore, wings, as resonant structures, need to deform appropriately at their actual flapping frequencies, and it is entirely possible that they may be tunable by active control of wing rigidity.

In the last two decades, particularly stimulated by the biomimetic possibilities in the development of micro air vehicles, there has been a great increase in interest in the structure, properties and functioning of the wings of certain groups: hawkmoths [25,26,29,36], locusts [23,37,38,39], hoverflies [40,41,42] and, above all, Odonata [43,44,45,46,47,48,49,50]; see [46,47,48] for reviews of the extensive literature, in which modelling has played an increasingly important role. Models have long been valuable in understanding wing functioning, and Wootton et al. [24] identified a logical sequence from conceptual though physical and analytical models to increasingly sophisticated computational simulations of individual species.

Each stage in this sequence has both advantages and limitations. Computational models now rightly dominate the literature, but they are vulnerable to incorrect initial assumptions, and, historically, some of the most useful information has come from simple physical models, based on direct observation of insects in flight and simple manipulation of wings. These are easy and quick to construct and have allowed the swift investigation and testing of a range of observed phenomena in a broad range of insects, in some cases giving direction to analytical and computational modelling of complete wings or wing components [3,12,23,24,38,47,49,50,51,52,53,54,55].

In 1999, I further discussed wing design, deformation and control in the wider context of invertebrate paraxial locomotory appendages, and illustrated how the principles underlying the in-flight deformation of many wings can be learned as a first approximation by modelling them as simple shells; see [55] for a wider range of references to research since 1981. The present paper uses physical models of this kind to extend the discussion by exploring how aspects of the geometry of the wings may affect their deformations in flight and to suggest how these may in theory be actively influenced and controlled remotely from the axilla. It will focus primarily on species that have either been specifically investigated or for which good photographic information is available. A selection of highspeed photographs by Stephen Dalton and John Brackenbury, some but not all previously published, are included with the authors’ generous permission.

I am not concerned here with the shape changes in the expanded anal fans of the hindwings of Orthoptera, Dictyoptera and some other orders (23, 37, 38, 39, 52), or with the flight deformations in Coleoptera hindwings, which are strongly influenced by the flexible lines by which these fold up at rest [17]. The emphasis is on mechanisms involving some transverse bending, particularly, but not exclusively, in forewings. Hindwings also deform in many groups, depending on their relative length and on the presence or absence of wing coupling. The latter also influences the nature of forewing deformation—compare the Trichoptera in Figure 2.

### 1.1. Rigidity, Flexibility and Active Control

In typical flight-adapted wings, certain areas are clearly adapted for rigid support, with thick veins, high relief and sometimes thickened membrane. These are generally in the proximal part of the wing and along the more anterior veins. Posterior support, necessary to prevent the wing from pitching into the airflow, is in the forewings and many hindwings of Neoptera generally provided by a rigid clavus, or in many Diptera by automatic mechanisms that lower the trailing edge in response to aerodynamic loading, a situation which is also characteristic of Odonata [12,15,50]. The forewings of Ephemeroptera have no clear clavus, but the anal area provides similar posterior stiffening.

In most insects, the profile of the wing base can alter by hinge-wise bending along specific longitudinal flexion lines [1,2,4,5], of which the most important and widespread are the claval flexion line and the median flexion line—the “remigial furrow” of Martynov [57] and Grodnitsky and Morozov [8]. Basal profile change is a frequent component of the active torsion of the whole wing during the stroke cycle and is the only way in which thoracic muscles can directly deform the wing.

All other deformations are passive responses to aerodynamic, inertial and occasional impact forces, and they tend to be concentrated in more distal areas of the wing, where the relief is flatter, the longitudinal veins are more slender, even sometimes absent, and cross-veins are relatively thin and flexible. These areas are sometimes clearly delineated by a visible transverse flexion line, marked by local areas of thinning of membrane and veins—“thyridia”—or by points or lines of soft cuticle that interrupt the veins themselves.

### 1.2. The Functions of Bending

Typical wings are thin, springy plates, stiffened by tubular veins, whose mass and thickness diminish along the span. Bending is often a simple response to the inertial forces as the wings decelerate at stroke reversal. Importantly, they only significantly flex ventrally; sometimes around the bottom of the stroke, followed by a sharp straightening, and sometimes throughout most of the upstroke. Dorsal bending is normally slight or absent, though long wings can sometimes flex alarmingly in response to gusts of wind or in extreme accelerations. Otherwise, the principal function of deformation is aerodynamic optimisation: to create necessary force asymmetry between the downstroke and the upstroke, or to generate bursts of unsteady lift.

The shape of the downstroke is fairly consistent: the wing is extended and pronated, usually slightly cambered, with a degree of spanwise twist—“washout”, the ideal situation for generating steady lift. Upstroke deformations can be far greater. In some cases, they merely serve to “feather” the wing by reducing its effective area or its angle of attack, so minimising adverse aerodynamic force, but many insects need to develop usefully directed lift throughout the stroke cycle. For this, passive torsion within the span is usually crucial. In Odonata, in many Diptera and in some other insects with uncoupled wings, most of the remigium can swing across like a sail around the supporting anterior veins, but in many other insects—particularly those with coupled wings, like Hemiptera, Hymenoptera, many Lepidoptera and some Trichoptera (Figure 2), or those with a long clavus—torsion is concentrated more distally and is facilitated by a degree of ventral bending, often accompanied by a reversal of camber from dorsally convex to dorsally concave. This brings the distal part of the wing into a favourable angle of attack and suitable profile for generating usefully directed force in the translational part of the stroke, and the dynamic process of changing shape can probably create valuable unsteady lift around stroke reversal.

This paper will use models to investigate the relationships between bending, torsion and camber in wings of this kind, including some Ephemeroptera, Hemiptera, Plecoptera, Megaloptera, Mecoptera and Hymenoptera. These three aspects of deformation are intimately connected. Flexural and torsional rigidity are affected by relief, which in wings can take the form of camber, corrugation or a combination of the two. Whereas a flat plate, or a relatively flat corrugated plate, is equally flexible to dorsal and ventral bending, camber in a thin plate imposes bending asymmetry, as a force applied to the concave side tends to increase the height of the section and hence its rigidity, while a force on the convex side causes the sides to buckle outwards and the section to flatten—an effect familiar to anyone who has used an extending steel ruler [1]. The supporting areas of wings commonly have a degree of built-in camber, ensuring that wing flexion is always ventral. Where the aerodynamic and inertial forces are centred behind the torsional axis, a cambered wing is also asymmetric in twisting, far more resistant to pronation than to supination [51,53] Both these properties are appropriate to the upstroke and are often crucial in determining the shape and attitude of the distal, most aerodynamically effective part of the wing. Under aerodynamic loading, the deformable area of the wing often assumes a cambered section, dorsally convex in the downstroke but concave in the upstroke, and this reversal of camber is also related to aspects of the wing’s geometry, as we shall see.

### 1.3. Modelling Insect Wing Deformation

For the purpose of modelling, I am distinguishing three types of support.

#### 1.3.1. Corrugation

Ephemeroptera and Odonata have fully corrugated wings, with all main vein stems diverging from close to the wing base and alternately occupying the crests and troughs of a fluted structure. Odonate wings do not bend significantly, but in several families of Ephemeroptera, Edmunds and Traver [58] found “bullae”—patches of soft, flexible cuticle—aligned across the wing in three or four of the main concave veins of the forewings, and they correctly identified these as adaptations to ventral bending. Mayfly bullae and their alignment have recently been described in more detail [59].

*Ephemera* species (Figure 3) have bullae on four major longitudinal concave veins: the subcosta SCP, two branches of the posterior radius RP and the posterior media MP, in a nearly straight line across the wing. Brodsky [19] observed bending in the subimago of *Ephemera vulgata* in flight, though he did not see it in the imago. Four other mayfly families, with quite different flight behaviours, have no bullae. Images found online of the much photographed *Palingenia longicauda*, which does not have bullae, show that ventral flexion can occur in their absence; the bullae appear to be adaptations for sharp, small-radius bending, without damage to the veins.

#### 1.3.2. An M Section

The remigial supporting areas of Plecoptera, Megaloptera, Mecoptera, Trichoptera and many Lepidoptera and Diptera typically have two longitudinal concave troughs. The leading edge spar formed by the costa C, the subcosta SCP and the anterior radius RA is the first; it provides support as far as the point where the SCP ends as a separate vein. The second trough follows the median flexion line, close to the media M in most Plecoptera, *Sialis* (Megaloptera), *Panorpa* (Mecoptera), and most Trichoptera and Diptera. Lepidoptera vary greatly [5]. In Noctuidae, like the *Phlogophora* figured here, the median flexion line lies well anteriorly in the wing. Transverse bending occurs in some members of all these orders, often (except for Diptera) in both fore and hind wings. Wing deformation in Diptera also varies depending on proportions and on the presence or absence of one or more costal breaks and flexion lines [13].

The pattern of bending is strongly influenced by the length of SCP, which often terminates very short of the wing tip, so that the anterior concavity is flattened beyond. Here, and beyond the clavus which provides posterior support, the section is like a shallow letter M or an inverted W.

Figure 4 shows a selection of wings in these groups, together with some photographs and drawings demonstrating the deformations they undergo. The drawings indicate the main flexion lines.

A series of comparative investigations in Russia in the 1980s and 1990s have supplied valuable information on wing deformations in flight [2,5,6,7,8,19,21,60]. In all cases, the insects were tethered, so the kinematics may not necessarily reflect free flight, but they illustrate the deformations that the wings allowed. Brodsky [60] filmed *Isogenus nubecula* (Plecoptera) and took a series of high-speed photographs of *Sialis morio* (Megaloptera) [19]. Ivanov filmed *Rhyacophila nubile*, *Ceraclia senilis*, *Brachycentrus subnubilis* and *Arctopsyche ladogensis* (Trichoptera) [6], and Grodnitsky with colleagues filmed a range of Lepidoptera [5,8,21]. All showed a degree of ventral bending at the end of the downstroke. Figure 4b, of *Isogenus* immediately after the extreme point of transverse bending, shows a deep groove in the remigium proximal to the flexion, and the distal area is strongly supinated. Later frames from the same sequence show rapid straightening and completion of torsion early in the upstroke, and Brodsky’s images of *Sialis* and Ivanov’s of Trichoptera show relatively fast recovery, but a high-resolution photograph of *Sialis lutaria* in free flight by Dalton [62] (Figure 4d) shows flexion and reversed camber at an advanced upstroke stage. The same seems to be the case in his photograph of *Phlogophora meticulosa* (Figure 4f)*,* indicating that flexion is maintained throughout the half-stroke, as it appears in Ennos’ film of *Panorpa* (Figure 1) and in some of Grodnitsky’s moth images [21].

#### 1.3.3. Camber

The basal sections of the forewing remigium of most auchenorrhychous Homoptera have a cambered section. The membrane between the veins is often thickened, a condition that is more strongly developed in the hemielytra of Heteroptera. The camber sometimes continues into the more deformable, distal area; otherwise, this is flat. The clavus, which varies considerably in length, is typically strongly three-dimensional and rigid, and any bending happens at or beyond its apex. A median flexion line is probably frequently present, though not always obvious in Homoptera and detectable only by manipulation [20].

Cicadas (Figure 5a–c) have a particularly obvious transverse flexion line in the forewing, as do Tettigarctidae, Hylicidae and some Cixiidae and Psyllidae [20]. In each case, the line follows a curved path from a break in the costal margin to the end of the clavus, with the apex of the curve towards the wing base—significantly, as we shall see. A variety of other Homoptera show the parallel development of a transverse, straight alignment of cross-veins, presumably localising bending. Cicadas have no median flexion line. Photographs by John Brackenbury (Figure 5b,c) show different degrees of transverse bending and camber reversal in *Tibicina haematodes*.

The hemielytra of Heteroptera (Figure 6) show the clearest differentiation between supporting and deformable areas in any insect. Posterior support continues beyond the clavus as a sclerotised bar at the trailing edge of the remigium. Betts [9,10] found that ventral flexion in flight does not follow the line of the corium margin but takes place within the deformable membrane, along a straight line between the anterior end of the corium and the tip of the posterior sclerotised bar, and this is evident in Brackenbury’s photograph of *Palomena prasina* (Pentatomidae) (Figure 6b, from [16]). Several families of Heteroptera have an additional transverse flexion line within the corium: the cuneal fracture. In Miridae, at least, bending can occur at this point as well as at the tip of the corium, increasing the degree of distal supination (Figure 6c, from [16]).

A cambered wing base is also typical of Hymenoptera, where fusion of the stems of M and the anterior cubitus CUA has eliminated the usual difference in relief between the two veins. Brackenbury [18] has reviewed wing deformation in a range of Hymenoptera. Figure 7b,c, from [16], clearly show flexion, torsion and camber reversal in a wood wasp and an ichneumon, and a photograph of a vespid in [61] and various high speed video sequences which are available online indicate that these are widespread in the order. In coupled wings like these, flexion in the small hindwings is virtually absent, and in-span bending and torsion are restricted to the distal part of the forewing, beyond the coupling.

Examples of forewing M sections and cambered sections are shown in Figure 8. Note that both categories show an overall dorsally convex curvature, ensuring preferential resistance to dorsal bending. The M section wings are distinguished by the presence of a concave branch of the median vein (arrowed), with the median flexion line adjacent.

## 2. Materials and Methods

All models were made of card and paper. The variations in the rigidity and resilience of the different areas of the wing can be crudely replicated by varying the thickness of the materials. The models are simple to construct, and readers are encouraged to make and play with their own versions.

### 2.1. Corrugation

Model 1 (Figure 9a) was made from an A4 sheet of paper, with a density of 80 g/m^2^, and follows Edmunds and Traver [58] in representing the Ephemeroptera condition as a pleated paper fan with a transverse line of notches, simulating the bullae, cut in the concave pleats. With the base held, gentle downward force was applied beyond the bullae until yielding occurred.

### 2.2. The M Section

Models 2 and 3 (Figure 9b–d) were made from thin card, with a density of 175 g/m^2^, though density was not critical. Both measured 29.5 mm × 12 mm. Model 2 represented the M section alone, without anterior support from the leading edge spar or posterior support from the clavus. The sheet was longitudinally folded into three equal panels, and the centre panel was folded in half to form an M section. One end was held firmly by insertion into an expanded polystyrene block, representing the wing base, and downward force was applied to the distal end.

Model 3 had the same dimensions as Model 2, but a long triangular concave fold was added to each of the outer panels, corresponding to the leading edge spar and the clavus, to stiffen the proximal part of the model.

### 2.3. Camber

The wings were modelled as rectangles, measuring 29.5 mm × 11.5 mm, with a supporting base made of stiff card and a distal deformable area of standard printing paper, with a density of 80 g/m^2^ (Figure 10).

One flexion line AO, parallel to the long sides of the rectangle and corresponding to a median flexion line, was made by cutting partway through the depth of the card. Camber was adjusted experimentally by bending along this line. Its height was represented by the angle ε about the axis AO. The other flexion line, which could be transverse or oblique, was provided by the distal edge of the supporting card. In actual wings, this line is usually curved, but to simplify the geometry in the models, it was made of two straight lines, BO and OD, meeting the median flexion line at point O. This is an acceptable simplification: models with a curved flexion line behaved in exactly the same way.

These models therefore had three variables: the obliqueness of the transverse flexion line, measured by the angle ζ between the longitudinal axis and a straight line joining B and D; the angle BOD, as measured in the flat model; and ε. The first two were part of the model’s design, while the third could be manipulated.

In Model 4 (Figure 10a), BOD was straight, so angle BOD = 180° and ζ = 90°. In Model 5 (Figure 10b), BOD = 120° and ζ = 90°. In Model 6 (Figure 10c), BOD = 90° and ζ = 60°. In Model 7 (Figure 10d), the anterior supporting card extended to the end of the model. BOD was 90° and ζ 40°.

One more model, Model 8 (Figure 10e), was produced in order to investigate the specific wing conformation of some families of Heteroptera which have an extra transverse flexion line, the cuneal fracture, and a longitudinal flexion line well anterior to the mid line. The model was made using thinner card than in Models 4-8, as the supporting base needed some flexibility if the model was to function. This is of course closer to the situation in actual insects than the thick card used in the other models; the latter was chosen so that the camber could conveniently be measured as the angle ε.

## 3. Results

### 3.1. Corrugation: Model 1

Pressing on the dorsal surface caused the cut pleats to move dorsally and the fan to flatten and bend ventrally, creating an effective one-way hinge. When the fan was allowed to expand laterally, the model was stable only in the unflexed state. In the actual wing, the veins on the ridges and the stiffness of the convex pleats would bring about an elastic return to the unbent state. If expansion was prevented, the fan buckled irreversibly.

### 3.2. The M Shaped Section, Models 2 and 3

Model 2. Moderate pressure applied to the dorsal side caused the concave ridge to click abruptly upward into the plane of the convex ridges, forming a sharp hinge in the concave ridge, with only minimal curvature in the convex ridges and momentary slight lateral elastic overall expansion, which recovered as the new position was reached; the model was bistable. Further pressure reached a threshold at which the sides of the model buckled outwards and the section underwent catastrophic, unstable bending, returning elastically to the intermediate position when pressure was released.

Model 3. Moderate pressure applied to the dorsal side caused the concave ridge to click upwards, as in Model 2. The extra anterior and posterior folds extended beyond the resulting hinge. A shallow v-shaped flexion line developed between the apices of the extra folds and the hinge. When the dorsal side was pressed harder, the extra folds prevented lateral buckling, and the model was able to undergo appreciably greater stable flexion (Figure 9d). In supplementary models in which the extra folds were shorter, overall bending did occur beyond their apices.

### 3.3. Cambered Sections, Models 4–9

When flat, and ε = 180°, all models responded equally to forces applied to the upper and lower surfaces, but as soon as slight camber was introduced, they bent only ventrally. Figure 10 illustrates what happened to each model when camber was applied to the card component, slightly reducing ε, and a downward bending force was applied by finger to the flexible paper component distally to the transverse flexion line. The same deformations could be induced by drag forces if the models were flapped.

When camber was applied to the base, Model 4 (Figure 10a), where angle BOD = 180° and ζ = 90°, was stable in only one position, with a positive camber in the paper component. A downward force on the flexible area bent it ventrally, but it returned elastically as soon as the force was released. Other models, not illustrated, in which BOD was 180° and ζ was acute, were also monostable.

All the other models, Models 5–7 (Figure 10b–d), where angle BOD < 180°, had two stable positions: straight, with a positive camber in the paper component, and deflected, with a negative camber. They could be snapped from one position to the other by downward and upward finger pressure. Model 5 simply bent, but in Models 6 and 7, the paper component twisted as well as bent. Other models, not illustrated, showed that the ratio of torsion to bending increases as ζ decreases, and this reached an extreme in Model 8, where the anterior support extended to the end of the model and there was no overall bending.

Models, again not illustrated, where ζ was constant but angle BOD varied, showed that the magnitude of bending and torsion at a given value of ε increased as BOD increased. The geometry here is essentially the same as that described by Haas and Wootton [54] in a practical and theoretical analysis of the mechanisms involved in the folding of the hind wings of beetles and some blaberid cockroaches, and the analytical model which they derived can be applied to the present problem. For this reason, I have used the same letters for points and angles as appear in their paper.

In their model (Figure 11a,b), four fold lines, three of one sense (concave or convex) and one of the other, meet at a single point, the “origin” O. If the model is planar and is capable of being folded completely flat along these lines, opposite pairs of angles around the origin must each total 180°.

In Figure 10, Models 5, 6 and 7 can be seen to correspond to those in Figure 11. In these, when camber was applied by reducing ε, and the paper membrane depressed into the concave position the latter assumed a curved section, whose apex automatically assumed the position of the convex fold line OC in Figure 11a. We can redraw Figure 10c, Model 6, as Figure 11d, with the line of the apex of the curve represented by a line, OC. Figure 10c and Figure 11d are effectively Figure 11c upside down, with angle ε below the model, AO, BO and DO convex instead of concave, and BC concave.

Haas and Wootton [54] applied vector analysis to calculate the coordinates of point C: c (x), c(y), c(z), for any given value of ε, assuming the fold lines to be of equal length, equal to 1. Using the dot product, they derived three simultaneous equations:Cos α = c(x)*cos δ + c(y)*sin δ cos ε + c(z)* sin δ*sin ε
Cos β = c(x)*cos γ + c(y)*sin γ
1 = c(x)^2^ + c(y)^2^ + c(z)^2^

Comparing Models 4–7: experimenting by manipulation shows that for a given value of ε, the magnitude and speed of deflection increase and leverage decline with greater values of angle BOD. When BOD is large, a tiny increase in ε causes significant bending, as well as torsion if ζ is acute. In Model 4 (Figure 10a), with BOD = 180°, deflection is theoretically maximal, as the paper could fold back flat over the cardboard—but, in fact, increasing ε merely increases distal camber; there is no leverage to drive deflection.

Manipulating Model 8 showed that increasing the camber about the longitudinal flexion line stiffened anterodistal support, opposing bending at the cuneal fracture.

## 4. Discussion

The limitations of the models discussed here are self-evident. Insect wings are not rectangles, flexion lines are often not straight or angular, and paper and card do not replicate the gradations in stiffness and resilience of insect cuticle. Their justification lies in the fact that they mimic deformations that are *known* to happen in flight. They are developed by experimenting with materials until their behaviour when manipulated matches that observed in actual wings. They then provide an appropriate first method for examining the geometry and mechanics underlying wing deformations, and they serve to give some direction to future investigations.

Corrugation in wings provides rigidity to transverse bending, but allows compliance to deformation that is parallel to the ridges and channels, and also to torsion provided that any cross-veins are flexible or have flexible joints with the longitudinal veins, like those discovered in Odonata by Newman [3] and comprehensively mapped by Appel and Gorb [63]. Odonata wings show torsion and camber change between half-strokes but have no bending adaptations; any bending being large-radius elastic responses to extreme loads, with immediate recovery.

This is not so with Ephemeroptera. Flight in mayflies, though brief, is crucial to reproductive success, and there is every reason to suppose their wings to be highly adapted for aerodynamic efficiency in the competitive circumstances of mating and oviposition. Bullae are characteristic of the families that use vertical nuptial flights. Vertical flight with a horizontal stroke plane allows aerodynamic force symmetry between the half-strokes, but bending may be needed for force asymmetry in directional flight by the subimagines and females; more kinematic information is needed. Bending requires the wing to flatten, compressing the veins in the concave pleats, and the bullae, like the notches in Model 1, allow these to buckle into the plane of the ridge veins without damage, with the stiffness of the ridge veins driving the elastic return. The bullae are almost in a straight line across the wing, so that flexion will be unstable, and the wing will return elastically, driven by the stiffness of the ridge veins.

The same problem faces other groups that use high relief for rigidity but need to bend. Hemiptera and Hymenoptera tend to meet this by relatively sharp differentiation between the rigid supporting base of the remigium and a flatter distal area, using the properties of camber to limit bending to ventral only. Many other insects, particularly among Plecoptera and Holometabola, show more gradual diminution of relief along the span and allow bending across moderate relief by upward buckling. Here, this is true of only one vein and an adjacent flexion line but is similar in principle to the situation in Ephemeroptera and paralleled in Models 2 and 3. In these cases, thyridia often serve the same function as the bullae of mayflies, allowing local buckling without damage.

With both solutions, there seems to be a distinction between some situations, as shown in Brodsky’s film of *Isogenia*, where bending—sometimes extreme—takes place around stroke reversal, followed by rapid straightening and torsion in the early part of the upstroke, and others, where some flexion continues throughout the upstroke, usually combined with some torsion and camber reversal. Both are likely to have aerodynamic consequences. The sharp, angular acceleration in the former situation may create useful transient unsteady lift; the latter condition would give steady favourable lift throughout the translational part of the upstroke.

In the former case, monostable bending, as in Models 3 and 4, is acceptable, but in the second, bistability is useful, and a curved flexion line is common, as simulated in Models 5, 6 and 7 and visible in those of *Sialis*, *Phlogophora*, *Panorpa*, *Tibicina*, *Urocerus* and *Ophion* (Figure 5, Figure 6 and Figure 7). In these cases, bending can contribute to torsion, and an oblique flexion line becomes valuable—expressed as ζ in Models 6 and 7, where flexion and torsion are interdependent. The inclination ζ depends greatly on the relative lengths of the anterior and posterior supports—of SCP and the clavus (in Heteroptera, the secondary rigid extension). This is illustrated in the Heteroptera in Figure 6a, and in the difference between the fore and hind wings of *Panorpa* (Figure 1) [14]. The ratio of bending to torsion depends on the value of ζ. The extreme condition, with torsion only, occurs where the anterior support extends to the wing tip, as in Model 7, e.g., in Odonata, sphingid moths and many Diptera and Hymenoptera, although many flies and Hymenoptera have a costal break—two in some Diptera—which allow a degree of flexion [18]. This can also enhance torsion, and the same is true of the cuneal fracture in mirid bugs (Figure 6c). Costal breaks in many Diptera are unusually proximally situated, and ventral flexion and consequent torsion are sometimes extreme—for example, in the supremely kinematically versatile *Calliphora*, whose wing can flex at two points, namely at the end of the SCP and close to the base [1,64,65]. Ennos [65] has discussed the possible aerodynamic implications of this, suggesting that the option of ventral flexion may give extra control of the force vector in all planes and contribute to their remarkable manoeuvrability.

Very little change in the basal camber of Model 7 was needed to alter distal wing torsion significantly, and the same was true of both bending and torsion in Model 6. Model 8 is also significant. As Betts [9] showed, the cuneal fracture may offer the options of flexion there, or across the membrane, or both, and this could well be controllable by altering the basal section about the median flexion line.

The physical models described demonstrate some mechanisms by which insects could potentially remotely control the instantaneous rigidity and shape of their wings in flight—but do they? The basal section in many insects certainly alters during the stroke by hinge-wise bending along the claval flexion line, but it is not yet clear how often and to what extent flexion along the median flexion line is actively employed to influence distal shape and attitude in flight. Basal camber could in theory be modified by altering the timing and/or amplitude of shortening of the basalar and subalar muscles. These typically act antagonistically to pronate and supinate the wing respectively over the fulcrum of the pleural wing process; a reduction, phasic or tonic, in the shortening amplitude of one or both could potentially induce camber in part of the stroke, but it may not be as simple as this. In the well-documented case of locust forewings, which control the distal angle of attack by assuming a basal z-shaped profile in the upstroke by flexion about both the median and claval flexion lines [66,67], the basalar and subalar muscles apparently act together to pronate the wing in the downstroke, while the principal supinator is the flexor muscle [68]. Heteroptera, many of which have a clear median flexion line in the corium that would seem to make them excellent candidates for active section control, have no basalar muscles; the wing is pronated phasically by the indirect dorsal longitudinal muscle acting through the first and second axillary sclerites [9,69]. The subalar muscle could perhaps induce basal camber by shortening tonically over several stroke cycles, but this is pure conjecture.

Too little is still known about the precise operation of the basal direct muscles and axillary sclerites of most insects, and electrophysiological as well as morphological research will be necessary to determine whether in any particular case of stroke-by-stroke variation in wing shape is actively controlled and wing rigidity actively tuned.

Whether or not the insects exert active profile control, the mechanisms do have possible technical applications. Much recent work has gone into designing wings for micro air vehicles, but these have for the most part been relatively unsophisticated, utilising the wings’ flexibility but not attempting section control. I have suggested elsewhere how the principles explored in this paper could be used in an insect-based MAV, with minimal additional actuation [70].

## 5. Conclusions

Interest in the intricate, fascinating structure of insect wings has grown enormously in recent years, with the expansion of biomimetic engineering and the development of new micromorphological techniques and computational modelling. Understandably, the emphasis has been on a few species, predominantly Odonata and Diptera, with outstanding flight capabilities. The broader picture provided by comparative studies, and hence of interest to entomologists as well as engineers, has in general been lacking. This paper has attempted to show how simple, quickly built, physical models can continue to be useful in investigating aspects of wing design, in explaining parallel adaptations across the range of insect groups and by indicating directions for more sophisticated modelling.

## Figures and Tables

**Figure 1 insects-11-00446-f001:**
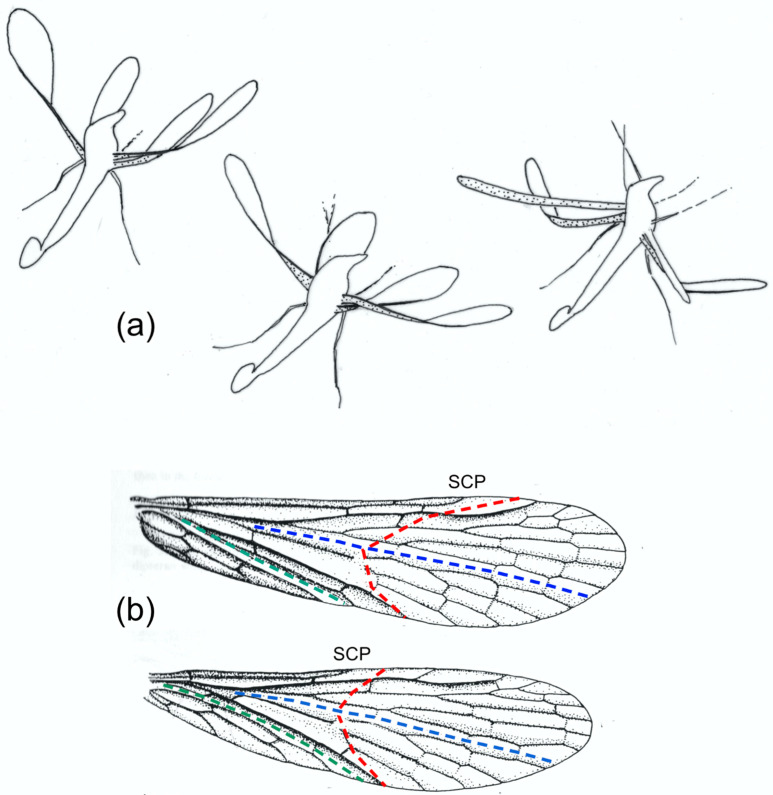
(**a**) Tracings of three frames from the same upstroke of *Panorpa germanica* from a high-speed film by A.R. Ennos. Note the very different bending modes of forewings and hindwings, reflecting the different lengths of the subcosta, SCP, and that flexion and torsion persist throughout the half-stroke. (**b**) Fore and hind wings of *Panorpa germanica.* Here, and in subsequent wing illustrations, the median flexion line is shown in blue, transverse flexion lines in red and the claval flexion line in green.

**Figure 2 insects-11-00446-f002:**
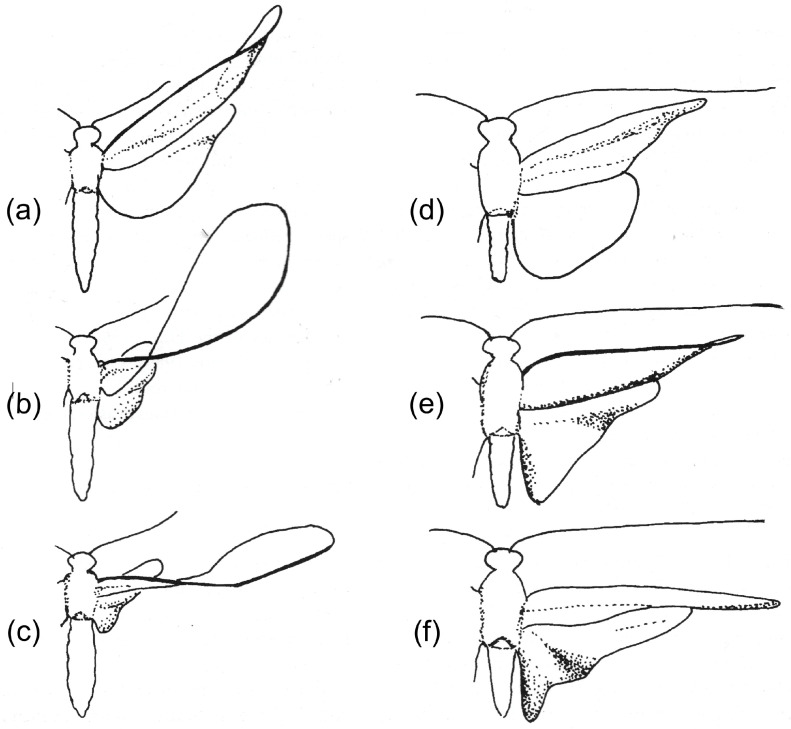
Tracings of successive frames from films of two Trichoptera in tethered flight, comparing a species with uncoupled wings (**a**–**c**) with a species in which the wings are coupled (**d**–**f**). From [56], redrawn after [6]. (**a**–**c**) *Rhyacophila nubile* (Rhyacophilidae). (**d**–**f**) *Ceraclia senilis* (Leptoceridae).

**Figure 3 insects-11-00446-f003:**
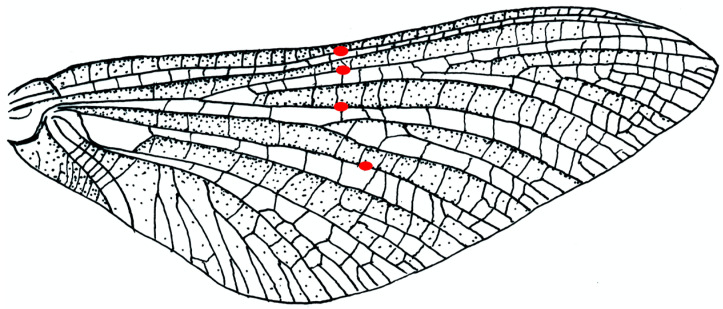
Forewing of *Ephemera vulgata* (Ephemeroptera). The positions of the bullae are shown by red spots.

**Figure 4 insects-11-00446-f004:**
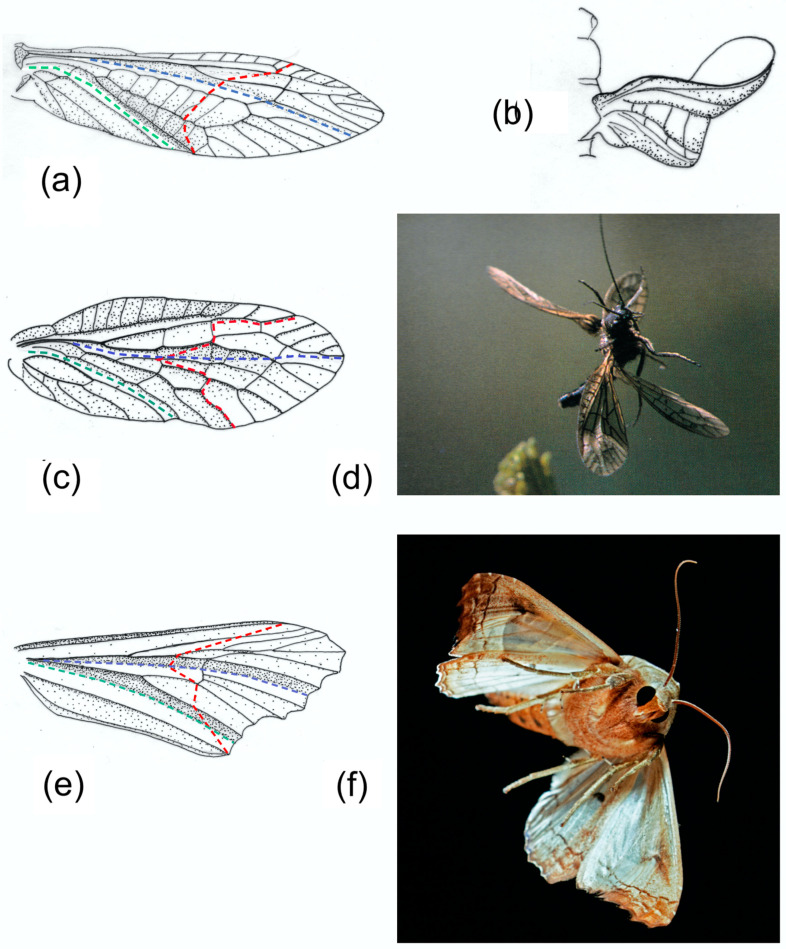
(**a**) *Isogenus nubecola* forewing. (**b**) Tracing of a frame of *I. nubecola* at the start of the upstroke in tethered flight. (**c**) *Sialis lutaria* forewing. (**d**) *S. lutaria* in late upstroke. (**e**) *Phlogophora meticulosa* forewing. (**f**) *P. meticulosa* in late upstroke. (**a**) and (**b**) are redrawn after Brodsky [60]. (**d**) and (**f**) are copyright Stephen Dalton. (**d**) has previously been published in [61], (**f**) in [62]. Red: transverse flexion line. Blue: median flexion line. Green: claval flexion line.

**Figure 5 insects-11-00446-f005:**
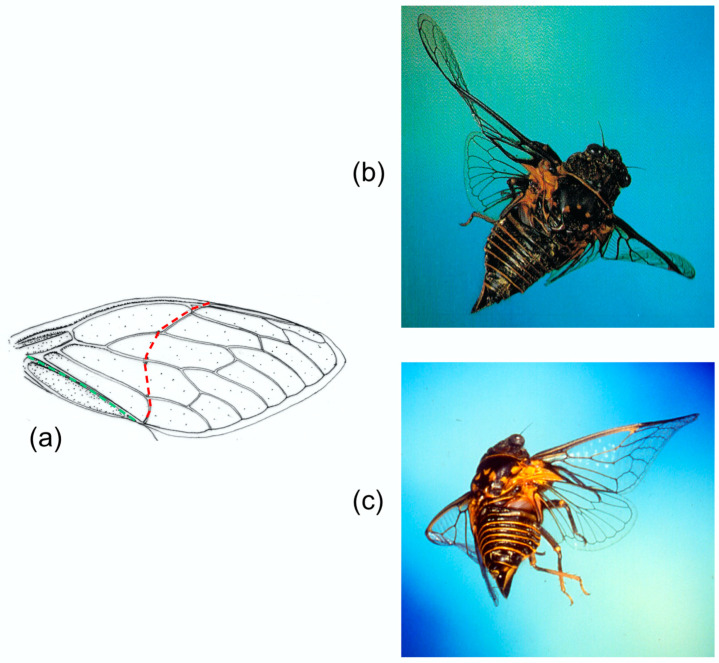
Upstroke deformation in *Tibicina haematodes* (Cicadidae). (**a**) The forewing. (**b**), (**c**) Two images of the upstroke. (**b**) Mid-upstroke, (**c**) early upstroke. (**b**) and (**c**) copyright John Brackenbury. (**b**) has previously been published in [16]. Red: transverse flexion line. Green: claval flexion line.

**Figure 6 insects-11-00446-f006:**
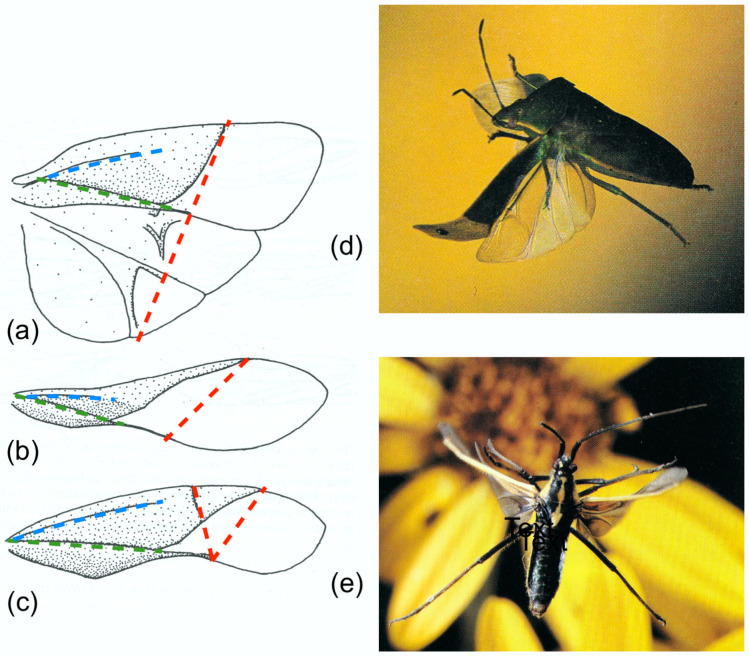
Wing proportions and upstroke deformation in Heteroptera. (**a**) Pentatomidae. (**b**) Alydidae. (**c**) Miridae. (**d**) *Palomena prasina* (Pentatomidae) early in the upstroke. (**e**) *Leptoterna dolabrata* (Miridae) in mid upstroke, showing flexion at the cuneal fracture, aiding supination. (**a**–**c**) From [61], (**d**) and (**e**) copyright John Brackenbury, previously published in [16]. Red: transverse flexion lines. Blue: median flexion line. Green: claval flexion line.

**Figure 7 insects-11-00446-f007:**
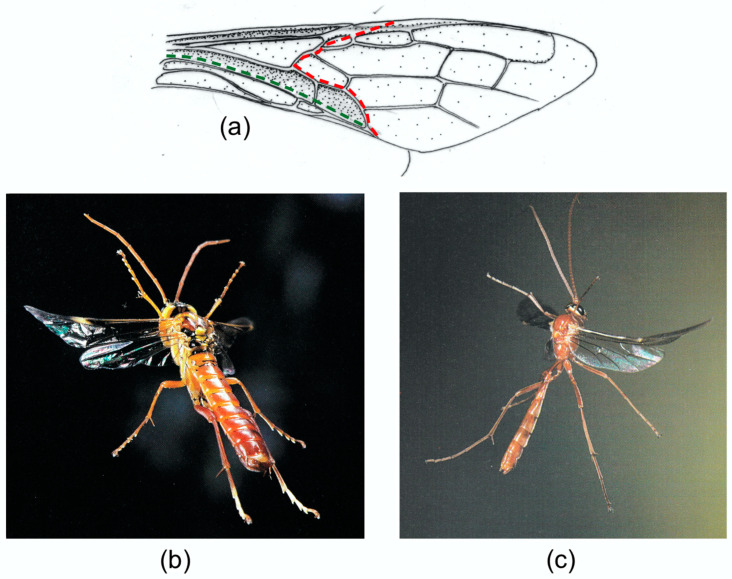
Upstroke deformation in Hymenoptera, showing flexion, torsion and camber reversal. (**a**) Forewing of *Urocerus gigas* (Siricidae). (**b**) *Urocerus gigas* male in mid upstroke. (**c**) *Ophion luteus* (Ichneumonidae) in early upstroke. (**b**) and (**c**) copyright John Brackenbury, previously published in [16]. Red: transverse flexion line. Blue: median flexion line. Green: claval flexion line.

**Figure 8 insects-11-00446-f008:**
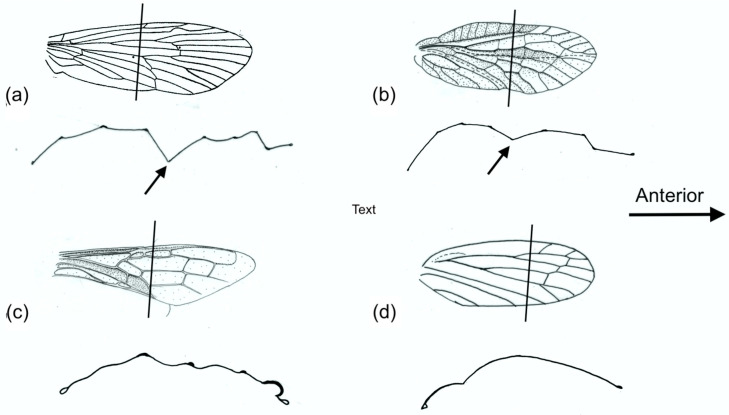
Forewing sections. (**a**) and (**b**) are M sections: (**a**) *Phrygania* (Trichoptera), (**b**) *Sialis* (Megaloptera). (**c**) and (**d**) are cambered sections: (**c**) *Urocerus* (Hymenoptera), (**d**) *Cercopis* (Homoptera). The lines indicate where the sections were cut.

**Figure 9 insects-11-00446-f009:**
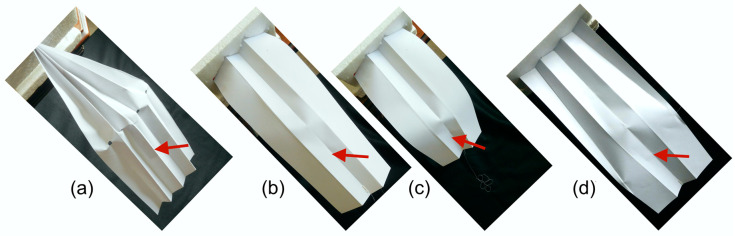
(**a**–**d**) Models 1–3 deforming. (**a**) Model 1. (**b**), (**c**) Model 2. In (**b**), the position is stable. In (**c**), under greater load, the sides are buckling outward, allowing unstable flexion that returns to position (**b**) when the load is removed. Flexion is directly transverse. (**d**) Model 3. The extra anterior and posterior folds delay unstable bending under load, and the flexion line is curved. The arrows indicate the approximate points and directions of the applied bending force.

**Figure 10 insects-11-00446-f010:**
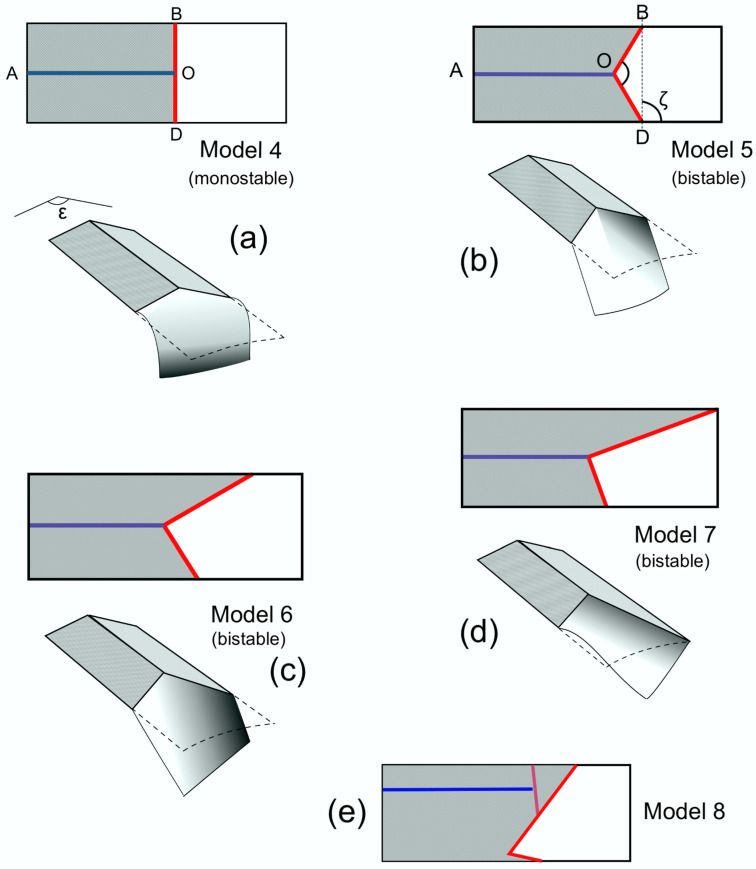
Models 4–8. (**a**) Model 4. (**b**) Model 5. (**c**) Model 6. (**d**) Model 7. (**e**) Model 8. The broken line is the outline in the unflexed state. Model 4 is stable only when unflexed; Models 5, 6 and 7 are bistable. Model 5 shows flexion only; Models 6 and 7 show torsion as well as bending. Model 8: explanation in the text. Red lines correspond to transverse flexion lines, blue lines to median flexion lines in wings.

**Figure 11 insects-11-00446-f011:**
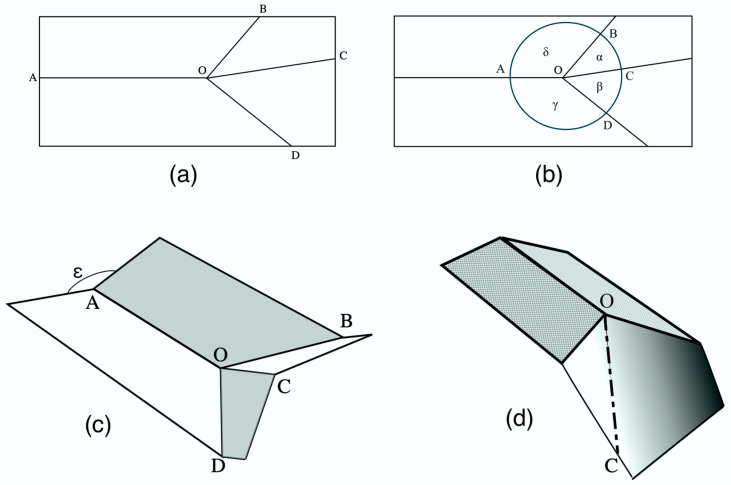
(**a**) A four-fold system, found extensively in Coleoptera hindwings. (**b**) The same, schematized for analysis by Haas and Wootton [54]. (**c**) The system partly folded. (**d**) Figure 9c modified, with a line representing the lowest axis of the curved membrane. Greek letters in (**b**) represent the planar angles around the origin, O.

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
