# Peer review of "The Geometry and Mechanics of Insect Wing Deformations in Flight: A Modelling Approach"

_insects, 2020, doi:10.3390/insects11070446_

Round 1

Reviewer 1 Report

The manuscript is surely interesting, but I'm not sure that the use of paper is   an adequate method to analyze the wing deformation during flight in insects.

I could suggest instead the use of a simulation software, in which the characters of the wings could be modified based on the differences detected in the various insect orders.

The folding can influence the shape modifications of the wings, but other factors must be tatken into account.

I could suggest to modify the manuscript into a review paper, giving a detailed account about the modification of the wings in the insects, and cutting the part about the paper folds.

Author Response

Thanks for the comments.  You reasonably question the validity of the models, but it is important to appreciate that these are not crude models of actual wings, but of mechanisms relevant to observed phenomena in flight. I justify their use in lines 67-73 (revised version), and acknowledge their limitations in lines 397-403. I am not qualified to carry out a simulation analysis. The modelling section is central to the paper, and I am not prepared to rewrite the paper as a review. An editorial decision; if this is insisted on I should prefer to withdraw the paper.

Reviewer 2 Report

The manuscript reviews the principles of wing deformation in insects, offering elaborate descriptions and explanations of flexing- and bending behaviors. A simple physical modeling approach demonstrates wing stiffness and deformation in wings made of paper and cardboard. The text is well-written in a book style thus the extensive knowledge of the author is not referenced at all places in the text. The article is somewhat between a review and an original paper because the author spent 9 of 17 pages on the introduction, while the data section is comparatively short and also uses data from a previously published manuscript (figure 10). The author might consider changing it to a review. In sum, I much like this contribution to the special issue of Insects and my comments are relatively minor.

Comments

P1 Why do you limit the analysis to the upstroke? This is written multiple times in the text but deformation may occur throughout the stroke cycle. My impression is that this limit does not hold for many of the descriptions later in the text (e.g. P1 L11, 24; P6 L199).

P1 L16. Active control of wing shape is still an actual issue. I was looking forward to that section in the text, but it was only mentioned in few lines at the very end of the discussion (P14 L429-433). I deeply encourage the author to extend this idea because there is little known about wing shape articulation.

P1 L13-15. Maybe this could be said in more simple words (what is a “shape of preferred lines”?)

P1 L27 Flexibility can be elastic and plastic.

P1 L38. Here and throughout the text, there are *many* “double spaces”. Please revise.

P2 L52. Here, a reference would be nice to have.

P2 L62 Add a comma at the end of the line.

P3 L95. Here and in all other subheaders: please remove the dot at the end.

P4 L106/107 See my above comment on active shape control.

P4 L119 What is “dorsal bending”?

P4 L121-122 This is a bit misleading because many insects with a horizontal stroke plane produce lift during up- and downstroke. Butterflies mainly produce thrust during the upstroke and lift during the downstroke.

P4 L128 Diptera …..with uncoupled wings? Please revise.

P4 L130 “…many other insects…” Please add some examples.

P4 L142. Here and at *many* other places of the text. It is often unclear in which direction the wing is pressed. In flight, pressure on the upper wing surface is suction pressure, pulling and not pressing on the surface. Throughout the manuscript I found it often difficult to know what “pressure” means? I suggest either to write “pressing on /pushing against the dorsal/ventral surface” or “pulling on the dorsal/ventral surface”, or saying that the wing is pushed downward during flight (lift is applied to dorsal and ventral surfaces).

Please be also more specific on the type of wing load, e.g. does the deformation pattern result from drag forces on the entire surface during flapping or from point loading. In the modeling section you could highlight the location of the point load by an arrow in the figures. This would also tell in which direction you press and that you press and not pull (suction pressure) or vice versa.

P4 L149 Here I don’t understand the term “curved section”.

P5 L179 Should be “in”.

P5 L181-186 Cross-sections would more clearly show the M-shape.

P8 L227 “shows”?

P8 L17 Do you here mean “Figure 7b and c from [16]”?

P9 L246. “order”.

P10, P11 L295. I think that you need to spend more writing on angle definitions. The problem is that in some cases the angles are only defined in a planar wing. For example, is BOD only the angle in a flat wing? If the wing is cambered (e.g. Fig. 9a, lower graph): does BOD change because the B,D, and O are projected into the 2D-plane? I also suggest to draw angles and Greek letters larger and more clearly. See also my comments to P13 L359-364 below.

P11 L297. “….=180° and Zeta=90°”.

P11 L327. Please try to keep the nomenclature for “pressing”. What are forces from “above and below”? Is that equal to pushing against the upper / lower surface? Or are the models flapped?

P12 L335-337. Could you refer to a figure and maybe visualize this in the figure?

P12 L355. How is point “C” defined?

P13 L359-364. This paragraph needs a larger revision. If you define point C in space such as C(x), C(y), C(z), we need to know what alpha, beta and gamma are? If C should be defined I would also write the equations such as: “C(x) = ……”, “C(y) = ……”, “C(z) = ……”, with x and y the horizontal and z the vertical.

P13 L365. What is “velocity ratio” and “mechanical advantage” in this wing?

P13 L391 Do you mean “aerodynamic” efficiency or “behavioral” efficiency? The long wings of mayflies produce quite a bit of inertial costs but have little induced drag. Not sure what dominates here. If you mean “behavioral” efficiency, a better term might be agility and maneuverbility.

P14 L428 I would greatly appreciate if you could extend the section on active wing shape control.

P16 There are some minor formatting issues in the references e.g. 36. and 38.

Figure legends

P2 L54. Here and all other legends: I suggest to give a full reference when ever possible for all pictures/figures published previously.

P7 L224. In all figure legends: please always start a new sentence with uppercase letter after “(a)” etc.

P6 L189-192. In all legends: it is confusing if you use the figure letters twice, e.g. “(d)”. I suggest to simply use “d” without brackets when you refer to “(d)”.

Figures 8 and 9. Please explain red and blue lines.

Figures

P3 L91-94. Here and all other figures (if appropriate): It would be nice to know the exact times at which the wings were captured within the stroke cycle, e.g. mid upstroke etc.

 P7 L224. This figure and in all other figures: please use the same font size in all figures.

 Figures 8 and 9. Could you write the model number next the graphs? This would help when you refer to the models without mentioning the figure.

Figures 8 and 9. I suggest to add some more information to the models. For eample: where do you press? In which direction do you press (arrow)? What are the two positions (shaded area, running ant area)? Are the positions stable or unstable? Please also make red and blue lines thicker.

Figures 8 and 9. For a comparison with real insect wings: consider to plot a real wing next to the model so that the reader can compare.

Figure 10c. Please label point “O”.

Author Response

Thank you for this careful examination of the paper, and for the comments which have identified some serious weaknesses in the original manuscript. Taking them point by point:

P 1. Deformation in the translational component of the downstroke is far less than in the upstroke of insects considered, and I describe it in lines 123-125 (revised version). Line 199 refers to the deceleration at the end of the downstroke and the transition into the upstroke.  I have modified line 123 to make this clear.

Line 16.  Very valid criticism - I was conscious that more was promised in the abstract that was delivered in the discussion.  The fact is that this is still unresolved, but I have completely rewritten and extended that component of the discussion(lines 453-473), and added four new references. 

Lines 13-15. Not sure why this isn't clear.  I have deleted ' preferred' in case it helps.

Line 27.  Certainly; but I think plastic deformation is likely to be minimal at insect flapping frequencies.

Line 38 et seq. A bad habit of mine.  All now eliminated, I hope.

P 2 line 52.  This is just my opinion. I doubt if I have been the only one to have suggested it, but I have no specific references.

Line 62. Done.

P 3 line 95 et seq.  Done.

P 4 lines 106 -7. Reply above.

Line 119, I could change this to 'upward bending' but I think dorsal is better, as it is a morphological rather than a spatial term.

Lines 121-122.  Certainly, but I don't imply otherwise. My point is that upstroke deformation introduces differences in in magnitude and direction of upstroke and downstroke forces, where this is appropriate. Force symmetry is necessary in horizontal stroke-plane hovering, and upstroke deformation other than torsion isn't always present.  Butterfly kinematics, incidentally, are very varied, and the classic work on nymphalids may not always apply.

Line 128.  Have added two words to clarify.

Line 130.  Examples added.

Line 142.  I am talking here about pressure, however applied, on a thin plate, not a wing.  The force on a wing will be a combination of inertia and the net aerodynamic force at that moment, the latter being a pressure differential between the two surfaces. I have altered 'pressure' to 'a force applied to', and hope this is OK.  I have also added arrows to the new figure 9 (previously 8) on new page 11.

Line 149.  Not sure why this is a problem.  I have changed 'curved' to 'cambered', which I was avoiding so as not to use the word twice in the same sentence, but this is only stylistic.

Reviewer 3 Report

This is a very nice paper, clearly reviewing previous research on insect wing design and then going on to show how the downward deflections of the wings of a variety of orders of insects can be abstracted into three different sorts of models, which are well described. There is no attempt to quantify anything but the models are ingenious and represent a first stage of understanding the process, which is more important than quantification, which could be done later.

I think there are two sets of changes which would help to make the paper even better. First of all, I had trouble understanding the diagrams, especially the identity of the blue, green and red lines. It's clear if you go back to figure 1, but I think that the explanation given there could do with also being included in the legends for the other figures. And it is not clear of the relevance of the claval furrow line. Why is it included? Something in the main text about this and maybe how it helps strengthen the inner wing or help folding or exactly how it operates. It might also be helpful to have a rough cross section of the M-profile wings to demonstrate how they are corrugated.

The other thing is that it might help to add a brief discussion about how these flexion mechanisms differ from the ones in the Diptera which are mostly, but not always (eg in the Tipulidae) much  more proximal. Also is there a phylogenetic pattern.

These are just suggestions, though, as it is a lovely, beautifully written paper.

Author Response

Many thanks for your time, kind remarks, and constructive suggestions. Taking them one by one

  1. I have added a colour key to the legends of each of the relevant figures, and also slightly increased the thickness of the coloured flexion lines.
  2. The claval flexion line is included in the diagrams for completeness. I do discuss their roles and that of the clavus in various places: lines 100 to 105, 187, 217-8, 223 and 465 in the revised version.
  3. I have included a new figure 8 showing a selection of M-shaped and cambered sections, and renumbered subsequent figures.
  4. I had deliberately kept clear of Diptera because I didn't want to get involved in the massive Drosophila literature, but have now included a few lines - 447-454, and two more references. 
  5. Phylogenetic pattern? I am keeping well clear!

Thanks again.